# A Year in the Life of U.S. Frontline Health Care Workers: Impact of COVID-19 on Weight Change, Physical Activity, Lifestyle Habits, and Psychological Factors

**DOI:** 10.3390/nu14224865

**Published:** 2022-11-17

**Authors:** Tracy L. Oliver, Rebecca Shenkman, Lisa K. Diewald, Paul W. Bernhardt, Mu-Hsun Chen, Caroline H. Moore, Peter G. Kaufmann

**Affiliations:** 1M. Louise Fitzpatrick College of Nursing, Villanova University, Villanova, PA 19085, USA; 2MacDonald Center for Obesity Prevention and Education, Villanova University, Villanova, PA 19085, USA; 3Department of Mathematics and Statistics, Villanova University, Villanova, PA 19085, USA

**Keywords:** COVID-19 pandemic, healthcare workers, lifestyle habits, weight change, eating patterns, physical activity, psychological factors

## Abstract

Healthcare workers (HCWs) experienced significantly higher burdens and life demands due to the COVID-19 pandemic. This study sought to assess the longitudinal effects among HCWs throughout the pandemic. Qualtrics surveys collected self-reported data on weight changes, eating patterns, physical activity (PA), and psychological factors with data organized by timepoints prior to the pandemic (PP0—prior to March 2020), baseline (M0—January 2021), month 6 (M6—July 2021), and month 12 (M12—January 2022). Eating patterns were negatively impacted at the M0, with reported increases in snacking/grazing (69.7%), fast food/take-out consumption (57.8%), and alcohol (48.8%). However, by M6 and M12 there were no statistically significant differences in eating patterns, suggesting that eating patterns normalized over time. Mean weight increased from PP0 to M0 by 2.99 pounds (*p* < 0.001, *n* = 226) and from PP0 to M6 by 2.12 pounds (*p* < 0.027, *n* = 146), though the difference in mean weight from PP0 to M12 was not statistically significant (*n* = 122). PA counts decreased from 8.00 sessions per week PP0 to 6.80 by M0 (*p* = 0.005) before jumping to 12.00 at M6 (*p* < 0.001) and 10.67 at M12 (*p* < 0.001). Psychological factors comparing M0 to M12 found statistically significant differences for depression (*p*-value = 0.018) and anxiety (*p*-value = 0.001), meaning depression and anxiety were initially increased but improved by M12. Additionally, higher scores on depression and insomnia scales were associated with lower PA levels. These overall results imply that the COVID-19 pandemic had immediate effects on the eating patterns, weight changes, PA, and psychological factors of HCWs; however, routines and lifestyle habits appeared to have normalized one year later.

## 1. Introduction

The coronavirus 2019 (COVID-19) pandemic resulted in a global health crisis causing major disruptions in nearly all aspects of daily life, inducing physical and mental strain, halting global economies, and altering lifestyles [1,2,3,4]. In the spring of 2020, the United States implemented mitigation strategies, such as a national stay-at-home order, social distancing restrictions, and the closure of non-essential businesses, to limit disease transmission [3,4,5]. By early January 2021, the COVID-19 death toll exceeded 400,000 and U.S. cases surpassed 20 million [6,7]. Throughout 2021, vaccination availability and eligibility expanded as public health officials coped with the emergence of new strains, requiring additional mitigation strategies and treatment [8]. Throughout the pandemic, health care personnel, including nurses and healthcare workers (HCWs), continued working to provide patient care. For many, the associated risk of infection increased job-related stress and the potential for emotional and physical health consequences [2,9,10,11,12,13,14,15,16].

Stressful life events can result in a variety of health effects, including, but not limited to, changes in eating behavior, diet quality, physical activity (PA) habits, and alcohol abuse [2,4,17,18,19]. In addition, the association of weight gain with psychological stress related to work and life constraints has been well documented [1,20,21]. Stress and ineffective or nonexistent coping mechanisms are frequently associated with increased consumption of less-nutritive but highly palatable “comfort foods” that are typically energy-dense and high in sugar and fat [2,22,23]. The emergence of COVID-19 had a global impact on lifestyle habits and daily diets among adults, as noted by a recent systematic review of 23 longitudinal studies from 12 countries that found an overall increase in snacking, consumption of ultra-processed foods, and alcohol, and a decreased consumption of fruits, vegetables, and fresh food during the pandemic [24]. Excessive intake and snacking of unhealthy foods and alcohol can cause weight gain, increasing the risk of heart disease, hypertension, and diabetes mellitus [25].

Similarly, disruptions in PA routines during the pandemic were frequently reported. Stay-at-home orders meant limited or no access to a gym and outside PA regimens [1,2,3]. HCWs, namely nurses, were particularly affected due to limited free time for exercise and other self-care measures associated with demanding working conditions and disruptions in shift and shift length [26,27]. In a cross-sectional study of 264 nurses surveyed on healthy lifestyle behaviors, 65.9% reported a reduction in PA during the pandemic [27]. With reduced energy expenditure from PA, weight gain was an unintended consequence for some. Zachary et al. reported a statistically significant correlation between reduced PA levels and increased weight gain [28]. Other studies captured steady weight gain over the course of spring 2020 [29] or varying amounts of weight gain and PA in the general population during the pandemic [5]. Research indicates that decreased PA habits resulting from the pandemic may continue for some time even after restrictions are lifted, which may have long-term implications for overall health [3].

In addition to food choice, weight, and PA alterations, many individuals also experienced psychological changes. The impact of the pandemic on the mental health and lifestyle of nurses and frontline HCWs includes sleep disturbances and increases in distress, depression, anxiety, and fatigue [11,27,30,31,32]. Poorer sleep quality and self-reported rates of insomnia were significantly increased in frontline HCWs, with 75% reporting poorer sleep quality and 57% reporting new-onset insomnia [31,33]. HCWs also had statistically significantly higher rates of sleepwalking, sleep terrors, and nightmares [31]. Multiple systematic reviews reported the psychological impact of COVID-19 and other viral epidemics, finding increased mental health problems, particularly among female nurses, and overall increased trauma or stress-related disorders, depression, and anxiety [6,14,34]. Individuals coping with increased anxiety and depression, particularly frontline HCWs, are at high risk of developing unhealthy eating patterns and changes in physical activity in response to stress [4].

While other recently published literature captures similar metrics of weight change, eating patterns, PA, and psychological impact related to COVID-19 [5,24,29,35], few focus solely on the HCW population in the United States. Even fewer are longitudinal that extend to one year. This special population had unique experiences which limit their extrapolation from other published literature. This study aimed to provide information regarding the short- and long-term impact on weight changes, eating patterns, physical activity, and psychological factors among frontline HCWs in the United States. Therefore, research questions (RQ) were proposed to discern the pandemic’s impact over four time points within a projected two-year study. They included the following:

RQ1: Did the COVID-19 pandemic have any effect on eating patterns?

RQ2: Did the COVID-19 pandemic impact weight and was baseline BMI predictive of any weight changes?

RQ3: Did the COVID-19 pandemic impact physical activity levels, and were there any associations related to weight changes or psychological factors?

RQ4: Did the COVID-19 pandemic impact psychological factors, and were there any associations related to eating patterns, weight changes, or physical activity?

## 2. Materials and Methods

### 2.1. Study Design

The CHAMPS Lifestyle Study (CHAMPS-LS) is a two-year ancillary study to the longitudinal COVID-19 Study of Healthcare and Support Personnel (CHAMPS; NCT04370821), considered the parent study. CHAMPS-LS participants were recruited from a national registry of essential healthcare workers, defined as adult (age 18 or older) healthcare personnel, support personnel, and first responders working in any healthcare facility or the community and involved in supporting the care of COVID-19 patients [36]. This paper reports on the pre-pandemic data (PP0—prior to March 2020), baseline (M0—January 2021, first CHAMPS-LS questionnaire deployed), month 6 (M6—July 2021), and month 12 (M12—January 2022) time points to assess weight change, PA, psychological factors and eating patterns over time in the context of the COVID-19 crisis among this population. The study was approved by the authors’ University Institutional Review Board.

### 2.2. Participants

Those who joined the parent CHAMPS Registry during May and June 2020, approximately at the peak of the first phase of the pandemic (*n* = 801), were invited by email on 5 January 2021, to participate in the CHAMPS-Lifestyle Study. Of these, 241 gave consent and gained access to the M0 CHAMPS-Lifestyle Study questionnaire, of which 234 responses were used for analysis (removing incomplete responses, duplicate responses, or pregnancy). Additionally, 154 of these participants completed the M6 data collection, and 128 completed the M12 data collection. No compensation was provided for participation.

### 2.3. Measures

Online questionnaires containing closed and open-ended questions were administered by the Qualtrics survey creation and distribution platform [37]. The questionnaires included inquiries about general demographics (such as height, weight, gender, race, employment status, etc.) and questions regarding changes in eating patterns, such as shopping and dining out or eating take-out food, since the start of the pandemic. The questionnaires also included queries about intentional and unintentional weight gain or loss. Eating pattern questions included changes in appetite, fruit and vegetable consumption, alcohol intake, portion sizes before and during the height of the pandemic, and intake of caffeinated and/or sugar-sweetened beverages and meals purchased outside the home. In addition, consumption of commercially prepared food and food purchasing habits were assessed. Physical activity was assessed using total exercise sessions (counts) and the Godin–Shephard Leisure Time Physical Activity Scale (GSLTPAQ) [38]. This paper also incorporated data on depressive symptoms, anxiety, and insomnia from the parent CHAMPS study collected from May to June 2020 and M12. Depression was assessed using the Patient Health Questionnaire (PHQ-2), one of the most used brief screens with adult populations, with good sensitivity and specificity for detecting major depression. Anxiety severity was assessed using the General Anxiety Disorder GAD-7 screening tool, shown to have strong criterion validity for identifying probable cases of GAD. Insomnia was measured using the Insomnia Severity Index (ISI), a reliable and valid instrument to quantify perceived insomnia severity [39,40,41].

### 2.4. Analytic Strategy

Statistical analyses were carried out using R 4.1.3 (R Development Core Team, Vienna, Austria), and significance was set at *p* < 0.05 [42]. The study population’s descriptive statistics summarized baseline data on weight, body mass index (BMI), eating patterns, PA, food shopping, and alcohol consumption behaviors. Paired *t*-tests and one-way ANOVA were conducted to determine whether there were differences in weight, BMI, and PA changes across time points. In addition, one-way ANOVA tests were conducted to determine whether differences in depression (PHQ-2), anxiety (GAD-7), and insomnia (ISI), symptoms existed and these differences were then compared across time points and by weight change groups (lost weight (<5 pounds), gained weight (>5 pounds), and little change (remained within ±5 pounds of body weight)). Tukey’s multiple comparisons were used when conducting ANOVA tests to control for multiple testing, and log and square-root transformations to normality were applied as necessary to reasonably meet model residual assumptions. Longitudinal linear mixed models were used to identify relationships with weight and PA over time, while multiple linear regression models were used to study relationships with the PHQ-2, GAD-7, and ISI at M12.

## 3. Results

### 3.1. Descriptive Analysis

Of the 234 responses, a total of 86.8% of the respondents were female (*n* = 203), the average age was 38.69 years (SD ± 12.00), and the majority were White/Non-Hispanic (91.9%, *n* = 215). The mean BMI among all participants prior to the pandemic (*n* = 226) was 26.58 kg/m^2^ ± 6.22, with BMI classifications as follows: Underweight (0.9%), Normal weight (46.0%), Overweight (31.0%), and Obesity (22.1%). Due to the low number of underweight individuals in the study, the normal and underweight groups were combined for most statistical analyses below. A bachelor’s degree was the most common degree possessed by respondents (48.9%, *n* = 114), followed by a master’s degree (32.2%, *n* = 75). The majority of HCWs were Registered Nurses (64.5%, *n* = 151), with other healthcare professionals and essential workers making up the remaining portion of the sample (35.5%, *n* = 83), and the majority of respondents were employed full-time (77.8%, *n* = 182). See Table A1 in Appendix A for more descriptive summaries of the data.

### 3.2. Eating Pattern Changes Related to the COVID-19 Pandemic

At the height of the pandemic, those in the study were more likely to report negative changes in eating patterns, with a majority of individuals reporting comparing PP0 to M0, with the increases in snacking/grazing (69.7%), fast food/take-out consumption (57.8%), and alcohol (48.8%). Most respondents reported no changes in sweetened beverage (76.3%), fruits (70.1%), vegetables (71.6%), and food portion sizes (55.5%). With the exception of appetite and sweetened beverage intake, the distributions of decrease/no change/increase responses were statistically different at M6 than at M0, with eating patterns trending back in the more positive direction. There was an increase in the percent of individuals reporting a decrease in snacking, fast food, alcoholic beverage intake, caffeinated beverage intake, and portion size, while there was a corresponding decrease in those reporting an increase in those categories. Similarly, there was an increase in the percent of individuals reporting an increase in vegetable and fruit intake at M6 compared to M0, while there was a decrease in the percent reporting a decrease in vegetable and fruit intake. The distribution of decrease/no change/increase responses was not statistically different at M12 than at M6 for any of the eating pattern categories, indicating that perhaps eating patterns regularized again by M6. See Table 1 for the complete descriptive report.

### 3.3. Weight Changes Related to the COVID-19 Pandemic

At M0, 14.2% reported weight loss greater than five pounds, 46.5% reported no weight change, and 39.4% reported weight gain greater than five pounds since the onset of the pandemic (See Table 2). Paired *t*-tests were used to compare mean weight between measurement times, with statistically significant mean increases of 2.99 pounds from PP0 to M0 (*p*-value < 0.001, *n* = 226) and 2.12 pounds from PP0 to M6 (*p =* 0.027, *n* = 146). The mean weight change of 0.81 pounds from PP0 to M12 was not statistically significant (*p* = 0.548, *n* = 122). See Table 3 and Figure 1 for weight changes over time classified by BMI category. Mean weight changes from M0 to M6, M0 to M12, and M6 to M12 were not statistically significant (*p*-values of 0.5984, 0.2546, and 0.8183, respectively), though smaller sample sizes limited the statistical power. Key finding: Results indicate weight gain over the first ten months of the pandemic (from PP0 to M0) followed by a retreat toward the prior weight.

#### Weight Changes Associated with BMI Category

An ANOVA test and multiple comparisons were conducted to determine if an individual’s weight at PP0 was associated with whether they gained/lost weight at the height of the pandemic (considered baseline, M0), where weight changes at the height of the pandemic were divided into three groups: gained >5 pounds (*n =* 89), little weight change (*n* = 105), lost >5 pounds (*n =* 32). Findings indicated differences in mean weight prior to the pandemic amongst the three groups of pandemic weight change (*p* < 0.001, *n* = 226), with the pre-pandemic mean weight being highest for those losing >5 pounds, the next highest for those gaining >5 pounds, and the lowest for those with little weight change. Additionally, a chi-squared test of independence determined that there was an association between BMI at PP0 and pandemic weight change (*p <* 0.001). Compared to what would be expected if no association existed, standardized residuals indicated that (1) those with underweight or normal BMI were more likely to experience little weight change (residual = 6.3) and less likely to gain or lose weight (residual = −4.0, −3.4); (2) those with overweight BMI were less likely to experience little weight change (residual = −3.29) and more likely to have gained weight (residual = 3.2); and (3) those with an obesity BMI were less likely to experience little weight change (residual = −3.9) and more likely to have lost weight (residual = 3.4). Key findings: Starting weight was associated with how an individual’s weight changed at the height of the pandemic (M0). Furthermore, while normal and underweight individuals were less likely to have weight changes, overweight individuals were more likely to gain weight, while individuals with obesity were more likely to lose weight.

Finally, a longitudinal linear mixed model was fit to determine whether there were any significant differences in weight change patterns over time (PP0, M0, M6, M12) based on BMI category (normal/underweight, overweight, obesity). Again, no differences were found, confirming the visual evidence in Figure 1 that the trajectories are parallel.

### 3.4. Physical Activity Changes Related to the COVID-19 Pandemic

Changes in PA over the four time periods, as measured by mild, moderate, strenuous, and total exercise counts, as well as Godin Scale scores, were observed, with ANOVA tests demonstrating differences over the times for all methods of determining PA (*p* < 0.001 for all PA measurements). Specifically, multiple comparison tests showed that mean total PA counts between PP0 and M0 decreased by 15%, from 8.00 to 6.80 (*p* = 0.005) before increasing to 12.00 at M6 (*p* < 0.001 compared to PP0) and 10.67 at M12 (*p* < 0.001 compared to PP0). Mild, moderate, and strenuous PA counts followed the same trend, decreasing from PP0 to M0 and then increasing at M6 and M12 to levels above PP0, though the decrease from PP0 to M0 was not statistically significant for mild or moderate counts. Similarly, the Godin Scale score decreased by 18%, from a mean of 41.4 at PP0 to 33.9 at M0 (*p* < 0.001) before increasing to 61.64 at M6 (*p* < 0.001) and 54.4 at M12 (*p* < 0.001). See Table 4 for a complete set of means and standard deviations. Key findings: The PA activity levels changed throughout the pandemic, with total counts of exercise and Godin scores decreasing initially during the pandemic before increasing to levels higher than even before the pandemic. While counts came back down at M12, they were not statistically different from M6.

#### 3.4.1. Physical Activity Changes Associated with Weight Changes

To evaluate whether there was an association between PA and weight changes across the time points (PP0, M0, M6, M12), a longitudinal linear mixed model was fit with weight as the response and time and activity counts/Godin Scale scores as predictors. Weight was found to be related to both total PA counts and Godin Scale scores (*p* < 0.001 for both). Specifically, we would estimate that for every five additional physical exercise sessions per week, the weight is expected to decrease by an estimated 2.4 pounds. Similarly, for every ten unit increase in Godin Scale score, the weight is expected to decrease by an estimated 0.96 pounds.

It was also of interest to consider how PA changes at M0 (measured as the difference in PP0 and M0 PA counts or Godin Scale scores) were related to weight change at the height of the pandemic. A one-way ANOVA uncovered differences in both the mean PA count changes and mean Godin Scale score changes across the weight change categories (*p* < 0.001 for both). Specifically, the mean PA count for individuals in the group losing >5 pounds increased by 2.1 during the pandemic and was significantly greater than a 0.2 decrease in mean PA count in the group experiencing little weight change (*p* = 0.038), which, in turn, was significantly greater than the 3.6 decrease in mean PA count in the group gaining >5 pounds (*p* < 0.001). Similarly, the mean Godin Scale score for individuals in the group losing >5 pounds increased by 9.8 during the pandemic and was significantly greater than a 1.9 decrease in the mean Godin Scale score group with a minimal weight change (*p* = 0.047), which in turn was significantly greater than the 20.7 decrease in mean PA count in the group gaining >5 pounds (*p* < 0.001). Key findings: Individuals who lost weight during the height of the pandemic (M0) were more likely to have increased their weekly exercise in comparison to the other groups, while those who gained weight were more likely to have decreased their exercise. See Figure 2 for a boxplot of the Godin Scale scores by weight change group.

#### 3.4.2. Physical Activity Changes Associated with Psychological Factors

A longitudinal general linear model was fit to determine whether there is a relationship between depression (PHQ2 scores) at the height of the pandemic (measurements obtained from the CHAMPS parent study (May–June 2020)) and PA counts over time; both total weekly exercise counts and Godin Scale scores were found to be associated with PHQ2 (*p* < 0.001 for both; *n* = 226). Specifically, a one-level increase in PHQ2 was estimated to be associated with an expected decrease of 1.05 weekly exercise counts, while each one-level increase in PHQ2 is estimated to be associated with an expected decrease in the Godin Scale score of 5.53.

Next, a longitudinal general linear model was fit to determine whether there was a relationship between anxiety (GAD-7 scores) at the height of the pandemic and PA counts over time. Again, neither weekly exercise counts nor Godin Scale scores were found to be related to anxiety at the height of the pandemic (*p* = 0.310 and 0.471, respectively; *n* = 226).

Lastly, a longitudinal general linear model was fit to determine whether there was a relationship between insomnia (ISI scores) at the height of the pandemic and PA counts over time; both weekly exercise counts and Godin Scale scores were found to be associated with ISI (*p* < 0.011 and 0.012, respectively; *n* = 226). Specifically, a one-level increase in ISI was estimated to be associated with an expected decrease of 0.96 weekly exercise counts, while each one-level increase in ISI is estimated to be associated with an expected decrease in the Godin Scale score of 5.27.

### 3.5. Psychological Factors Related to the COVID-19 Pandemic

#### 3.5.1. Depression

Depression was measured using the two-item PHQ2, which asks about the frequency of depressed mood and anhedonia over the past two weeks. A score ranges from 0 to 6, with a score of 3 as the cut point when screening for depression (severe category) [39]. Data from the CHAMPS parent identified the presence of moderate depression in 27.4% of the sample and severe depression in 25.2% of the sample (see Table 5). A McNemar–Bowker test for paired categorical data was used to test whether the distribution of categories was different between M0 at the height of the pandemic and M12, with differences statistically different for PHQ-2 (*p*-value = 0.018). Specifically, improvements in depression were observed at the M12 follow-up, with a greater percentage of individuals in the “none” PHQ-2 categories.

A cumulative logit model was fit to determine whether there were any significant relationships between PHQ2 level and other lifestyle variables in the data set (fitting one predictor at a time). Each variable was an M12 measurement unless noted, and due to few responses above 3, the 3–6 scores were combined for the analyses. The following fitted relationships resulted in *p*-values > 0.05: weight, weight change, eating pattern changes (portion size changes, fast food consumption changes, fast food quantity changes, alcohol intake changes, and alcohol quantity changes). In addition, there was strong evidence of the following associations with higher PHQ2 at M12: lower total exercise count (*p* < 0.001), eating habits not returning to what they were PP0 to M6 (*p* = 0.006), worsened health change report from M6 to M12 (*p* < 0.001), higher GAD7 (*p* < 0.001), higher ISI (*p* < 0.001), lower Godin Scale score (*p* < 0.001). Given the number of tests conducted, the following fitted relationships only indicated moderate evidence of higher PHQ2: PA levels returning to what they were PP0 to M6 (*p* = 0.0204) and an increase in snacking habits (*p* = 0.028).

#### 3.5.2. Anxiety

Anxiety was measured using the GAD-7, a seven-item validated scale used as a tool for screening for generalized anxiety disorder, with a total score for the seven items ranging from 0 to 21, indicating minimal (0–4), mild (5–9), moderate (10–14), and severe anxiety (15–21) [40]. Data from the CHAMPS parent identified the presence of mild anxiety in 43.2% of the sample and moderate anxiety in 17.9% of the sample (see Table 5). A McNemar–Bowker test for paired categorical data was used to test whether the distribution of categories was different between M0 at the height of the pandemic and M12, with differences statistically different for anxiety (*p*-value = 0.001). Specifically, improvements in anxiety were observed at the M12 follow-up, with a greater percentage of individuals in the “none” GAD-7 categories.

A cumulative logit model was fit for GAD-7 anxiety level (minimal–severe) based on other lifestyle variables in the data set (fitting one predictor at a time). Each is an M12 measurement unless noted. The following fitted relationships resulted in *p*-values > 0.05: weight, weight change, eating pattern changes (portion size changes, fast food consumption changes, fast food quantity changes, alcohol intake changes, and alcohol quantity changes). There was strong evidence of the following associations with higher GAD-7: PA levels not returning to what they were PP0 to M6 (*p* = 0.003), decreased snacking (*p* = 0.007), eating habits not returning to what they were PP0 to M6 (*p* = 0.003), worsened health change report from M6 to M12 (*p* = 0.003), higher ISI levels (*p* < 0.001), and PHQ2 (as discussed above). The following fitted relationships only indicated moderate evidence of higher GAD-7: lower total exercise count (*p* = 0.023) and lower Godin score (*p*-value = 0.030).

#### 3.5.3. Insomnia

The ISI is a seven-item self-report measure that assesses symptoms of insomnia with individual item scores ranging from 0 to 4 and total scores ranging from 0 to 28 with four ordinal levels of insomnia severity: no clinically significant insomnia (0–7) and mild (8–14), moderate (15–21), and severe insomnia (22–28) [41]. Data from the CHAMPS parent identified the presence of subthreshold insomnia in 43.2% of the sample and moderate insomnia in 25.3% of the sample (see Table 5).

Next, a cumulative logit model was fit for ISI based on other lifestyle variables in the data set (fitting each predictor one at a time). Each is an M12 measurement unless noted. The following fitted relationships resulted in *p*-values > 0.05: weight, weight change, eating pattern changes (portion size changes, fast food consumption changes, fast food quantity changes, alcohol intake changes, alcohol quantity changes), and PA Godin Scale scores. In addition, there was strong evidence of the following associations with higher ISI: worsened health change report from M6 to M12 (*p* = 0.002), and higher M12 GAD-7 and M12 PHQ2 levels (as discussed previously). However, the following fitted relationships only indicated moderate evidence of higher ISI: lower total exercise count (*p* = 0.042). Lastly, it was noted that the distribution of categories for ISI was not statistically different between M0 and M12 (*p*-value = 0.737).

## 4. Discussion

To our knowledge, this study is the first to report on four time points within a projected two-year study designed to capture predictors of weight change, physical activity, psychological factors and eating behavior-related adaptations among HCWs in the United States during the COVID-19 pandemic. Studies to date that have investigated comparable measurements have been largely cross-sectional and during the initial months of the pandemic, and targeted the general adult population [43]. In addition, studies that are longitudinal have either not spanned the duration of this current study or have not focused specifically on HCWs, rendering this research a valuable and warranted addition to the literature library on this topic [24,35,44]. The longitudinal study design, when compared to similar research on this target population, is intended to detect developments, changes and/or reversals of these variables at both the group and the individual level, with COVID-19 being the change impetus. Identifying and monitoring health-related indicators during the pandemic is necessary to assess the physical and mental health changes in a population known to have experienced heightened job-related stress, potentially impacting emotional and physical health. The following discussion sections are organized to address each research question as stated in the introduction.

### 4.1. COVID-19 Impacts on Eating Patterns

Eating pattern changes were reported by HCWs in this study population. Half of the respondents (50.7%) reported that pandemic-related changes led to a disruption in meal consistency routine at work and/or at home. Meal timing alterations, such as delayed or skipped meals and late evening eating, may disrupt the circadian rhythm for some individuals, leading to an increase in obesity risk and unhealthy consequences [45,46]. While this study cannot establish causality between the potential effect of meal schedule consistency and weight control, our findings suggest and support that meal timing can have an impact on health and weight, similar to the quantity and quality of food we eat. While limited research has reported on eating behaviors specific to HCWs during COVID-19, eating behaviors among the general adult population worldwide have been studied. Systematic reviews on eating behaviors found comparable trends to this data, notably an increase in alcohol consumption, and an increase in snack consumption, and a general increase in appetite [24,43]. Additionally, our findings indicate that while there was an initial disruption in eating behaviors when comparing M0 to M6, the lack of these disruptions being sustained between M6 and M12 suggests that eating behaviors normalized again over time.

### 4.2. COVID-19 Impact on Weight Changes

In line with similar findings reported for the U.S. population over age 21 at large, 39% self-reported a weight gain of >5 pounds, and 14% reported weight loss of >5 pounds from PP0 to M0. This aligns comparably with reports of unintended weight gain and loss of 42% and 18%, respectively, reported by a U.S. adult population during a comparable timeframe [47]. In addition, a systematic scoping review on the impact of COVID-19 on weight management in healthy adults, where over half the studies were conducted in the European region, over 83% were cross-sectional descriptive studies, and over 72% reported data from April and May 2020, featured studies where 12.8% to 48.6% of study respondents reported weight gain and 13.9 to 19.4% reported weight loss (the amount of weight gain and weight loss varied across studies) [43]. Our sample also reported a mean weight increase of 2.99 pounds from PP0 to M0 and a mean weight increase of 2.12 pounds from PP0 to M6, which is higher compared to another study that tracked weight change from April–May 2020 to September–October 2020 among 764 adults between the ages of 18 and 75 and found a mean weight increase of 1.5 pounds [35]. Since our participants were HCWs experiencing high-stress levels and routine disturbance, it is not surprising to see higher weight gains. When looking specifically at the HCW population, studies from across the globe found comparable trends of weight gain during the first wave of the pandemic (considered winter/spring 2022) which parallels our M0 timepoint [48,49,50]. One such study that assessed the nutrition status of HCWs during the COVID-19 response period from March to April 2020 found 26.2% self-reported weight gain and 22.9% self-reported weight loss [50]. The research gap of longitudinal weight trends among HCWs during the pandemic is evident.

Additionally, this study’s longitudinal nature tracked weight change over time with results indicating that starting weight (or BMI) was associated with how an individual’s weight changed as reported during the height of the pandemic (M0). Individuals with lower or normal BMIs were more likely to experience little weight changes than individuals with a higher BMI, who were more likely to both lose and gain more. However, regardless of weight change from PP0 to M0, a slight weight gain persisted by M0 across all BMI groups before slowly waning. While BMI was associated with weight change at M0, there was no statistically significant evidence to suggest different changes in weight over time among the BMI groups. These findings may suggest that any initial differences in the initial BMI category were negligible by the time the height of the pandemic subsided, which could be due to many factors, including the possibility that those who gained weight wanted to lose it after the initial gain during the height of the pandemic, or a return to a more regular routine. This data trend parallels a recent U.S. study that shows mean weight gain among adults (noted, not HCWs specifically) during the COVID-19 pandemic may be smaller than originally thought at the beginning of the pandemic [51].

### 4.3. COVID-19 Impact on Physical Activity

Physical activity showed significant overall declines in both rate and intensity of physical activity as measured by the Godin–Shepard Leisure-Time Physical Activity Scale score when comparing levels prior to the pandemic with those during the height of the pandemic [38]. Several factors may explain the decline in physical activity rates, such as gyms being closed due to lockdown requirements and disruptions in routine. Notably, the statistically significant reduction in physical activity observed in this study, from 8.0 sessions per week prior to the pandemic to 6.80 sessions per week at the height of the pandemic (a 15% reduction in counts and an 18% decrease in Godin score), is similar to findings reporting a reduction in physical activity among HCWs during the pandemic (24). It should also be noted that despite the decline in physical activity among HCWs during this time, this population still exceeded the general population who achieves physical activity recommendations according to CDC reports (53.3% among US adults, whereas our study reported 72% as being active prior to the pandemic and 53.7% still being active at the height of the pandemic) [52].

Additionally, PA does seem to vary over time, and it appears that the total counts of PA and Godin Scale scores decreased during the pandemic period but then increased to higher levels as the height of the pandemic resolved and routines normalized. It appears that PA counts peaked at M6 and then came back down at M12; however, PA counts were not statistically different at M12 than M6. It should also be noted that both M6 and M12 PA values increased above the PA values at PP0 and M0, indicating an increase in PA since before and during the height of the pandemic. The analysis also indicated that changes in exercise level during the height of the pandemic were associated with weight changes at the height of the pandemic. Specifically, the average total exercise count and Godin score increased between PP0 and M0 among those who lost weight, stayed about the same for those whose weight did not change, and decreased for those who gained weight. It is possible that the mild increase in calorie expenditure from the increased PA helped with weight control, especially if accompanied by improvements in eating behaviors.

### 4.4. COVID-19 Impacts on Psychological Factors

It is clear that psychological factors were impacted during the pandemic and many sources have reported a pattern of increased anxiety and depression notably during the beginning months of the pandemic [53]. One systematic review estimated more than 53 million additional cases of major depressive disorders and 76 million additional cases of anxiety disorders globally in 2020 [54]. This study investigated the mental health burden of the pandemic on HCWs and, not surprisingly, showed negative health habits, such as less exercise and snacking changes, and generally worse feelings about one’s health, associated with higher levels of depression, anxiety, and insomnia. These results are comparable to studies of COVID-19-induced psychological stress among HCWs which demonstrated moderate and high levels of stress, anxiety, depression, sleep disturbances, and burnout [55,56,57,58,59]. Of interest, our results indicated no relation between BMI and depression, anxiety, or sleep variables at M12. Our study also indicated that despite many individuals reporting increased depression, anxiety, and insomnia at the height of the pandemic, by M12 there were improvements in these psychological factors, with a greater percentage of individuals in the “none” depression and anxiety categories. These findings follow trends to a systematic review and meta-analysis of longitudinal cohort studies examining changes in mental health among both non-clinical and clinical populations before versus during the pandemic that similarly found an increase in mental health symptoms soon after the pandemic outbreak, which then decreased and was comparable to pre-pandemic levels [53].

### 4.5. Next Steps

The importance of identifying and monitoring health-related indicators during the pandemic is necessary to assess the trajectories of weight and health habits. Additionally, exploring the data by BMI categories may provide new insight into stress responses or coping mechanisms related to individuals of different body weights, a topic that deserves further exploration and understanding. This study reinforces the importance of recognizing changes in healthy lifestyle behaviors, including eating patterns, physical activity, and weight changes, as well as monitoring the prevalence of sleep disturbances, stress, and anxiety brought on by stressful circumstances in the HCW population, which may increase the risk of chronic disease. Based on our study findings, the impact of the pandemic accentuated work-life and self-care challenges for some HCWs, worsening lifestyle behaviors and increasing psychological stressors. Prioritizing the provision of wellness support programs for health care workers to facilitate psychological support, resilience, and adaptability may help offset the impact of stress and pandemic-related disruptions on overall health and well-being. See Table A2 in Appendix A for a summary of key findings.

### 4.6. Limitations

There are several limitations of this study. We have a relatively homogenous sample, that is, a significant lack of job, racial, ethnic, and gender diversity; thus, generalizations to the larger population cannot be made. Our sample resulted in primarily full-time (77.7%), dayshift (58.2%), white (91.9%), female (86.8%) registered nurses (64.5%). This homogeneity likely underrepresented other experiences during the COVID-19 pandemic, such as those from individuals identifying as ethnic and/or racial minorities, those working night shifts, and frontline healthcare workers who are not nurses. Additionally, many of the measures collected were self-reported, whereas including objective measurements would enhance the confidence in study findings by reducing self-report bias. The authors also acknowledge that the pandemic is likely not the only contributing factor to the trends and changes reported. Additionally, while very little data was missing for the PP0 and M0 time points (data was usable for 97.1% of 241 individuals enrolling in study), there was a 34.2% dropout at M6 and a 45.3% dropout at M12. Furthermore, missing percentages were slightly higher for certain questions than others due to nonresponse. However, the proportion of individuals in various demographic categories remained relatively consistent across the time points, indicating that the main drawback of missing values may be in the decreased power to find statistically significant differences at the M6 and M12 time points.

## 5. Conclusions

This paper is the first to report on year 1 of a 2-year longitudinal study of weight changes, eating patterns, physical activity, and psychological factors among a specified group of HCWs. The findings contribute to the current body of growing evidence to best understand how pandemic-induced lifestyle disruptions shape health behaviors and weight change among HCWs. The ability to track our participants over the peak of the pandemic serves two purposes: (1) to identify modifiable risk factors and health indicators to inform public health programs and preventative efforts during the current crisis and potential future pandemics, and (2) to assess whether changes or lapses extend to a permanent cessation of those behaviors throughout the duration of the study. Thus far, based on these findings, it appears that despite many initial impacts related to negative eating patterns, weight gains, decreased physical activity, and negative psychological factors in response to the pandemic, most of these behaviors normalized or even improved over time.

## Figures and Tables

**Figure 1 nutrients-14-04865-f001:**
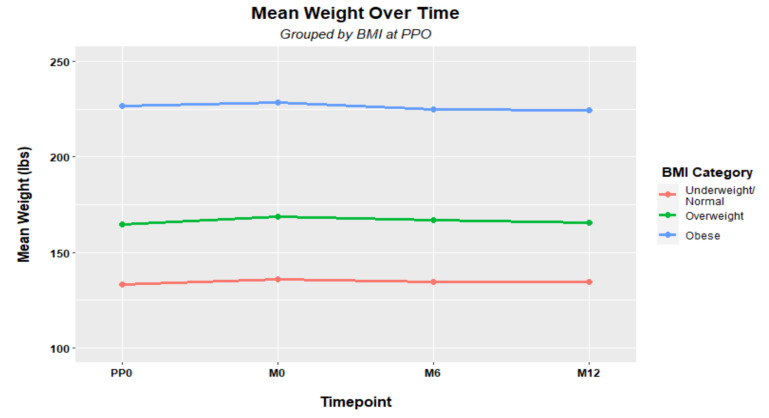
Mean weight over time by BMI category.

**Figure 2 nutrients-14-04865-f002:**
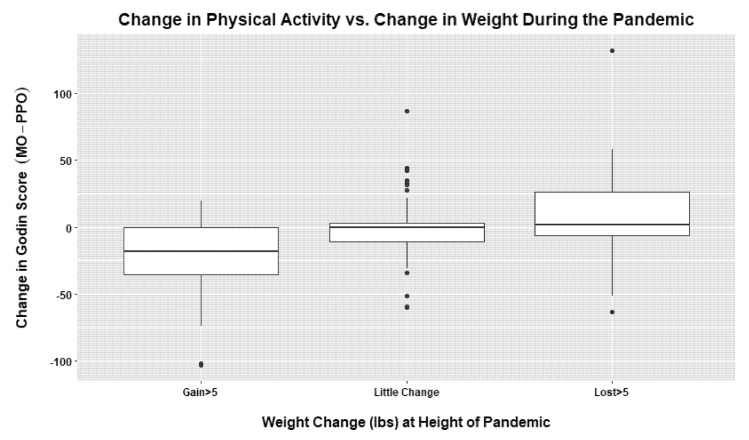
Change in physical activity vs. change in weight during pandemic based on Godin Scale scores (height of pandemic M0 vs. prior to pandemic PP0).

**Table 1 nutrients-14-04865-t001:** Eating pattern changes in consecutive survey time points (PP0 to M0, M0 to M6, M6 to M12); McNemar–Bowker test *p*-values for whether the current six-month distribution of decrease/no change/increase is different from previous distributions.

	PP0 to M0	M0 to M6	M6 to M12
Snacking/Grazing	*n* = 211	*n* = 148	*n* = 125
Less/Decrease	15 (7.1)	28 (18.9)	21 (22.4)
No Change	49 (23.2)	59 (39.8)	54 (47.2)
More/Increase	147 (69.7)	61 (41.2)	50 (48.8)
Change from previous?	-	*p*-val < 0.001	*p*-val = 0.953
Fast food or take-out consumption	*n* = 211	*n* = 148	*n* = 125
Less/Decrease	27 (12.8)	27 (18.2)	17 (13.6)
No Change	62 (29.4)	83 (56.1)	73 (58.4)
More/Increase	122 (57.8)	38 (25.7)	35 (28.0)
Change from previous?	-	*p*-val < 0.001	*p*-val = 0.733
Alcoholic beverage intake	*n* = 211	*n* = 148	*n* = 125
Less/Decrease	25 (11.8)	26 (17.6)	19 (15.2)
No Change	83 (39.3)	91 (61.5)	81 (64.8)
More/Increase	103 (48.8)	31 (20.9)	25 (20.0)
Change from previous?	-	*p*-val < 0.001	*p*-val = 0.991
Caffeinated beverage intake	*n* = 211	*n* = 148	*n* = 125
Less/Decrease	6 (2.8)	11 (7.4)	11 (8.8)
No Change	104 (49.2)	81 (54.7)	66 (52.8)
More/Increase	101 (47.9)	56 (37.8)	48 (38.4)
Change from previous?	-	*p*-val = 0.014	*p*-val = 0.666
Appetite	*n* = 211	*n* = 148	*n* = 125
Less/Decrease	32 (15.2)	25 (16.9)	20 (16.0)
No Change	94 (44.5)	85 (57.4)	84 (67.2)
More/Increase	85 (40.3)	38 (25.7)	21 (16.8)
Change from previous?	-	*p*-val = 0.063	*p*-val = 0.532
Food portion sizes	*n* = 211	*n* = 148	*n* = 125
Less/Decrease	16 (7.6)	32 (21.6)	24 (19.2)
No Change	117 (55.5)	89 (60.1)	82 (65.6)
More/Increase	78 (37.0)	27 (18.2)	19 (15.2)
Change from previous?	-	*p*-val < 0.001	*p*-val = 0.224
Sweetened beverage intake	*n* = 211	*n* = 148	*n* = 125
Less/Decrease	13 (6.2)	16 (10.8)	9 (7.2)
No Change	161 (76.3)	116 (78.4)	106 (84.8)
More/Increase	37 (17.5)	16 (10.8)	10 (8.0)
Change from previous?	-	*p*-val = 0.257	*p*-val = 0.435
Daily vegetable intake	*n* = 211	*n* = 148	*n* = 125
Less/Decrease	33 (15.6)	13 (8.7)	17 (13.6)
No Change	161 (71.6)	90 (60.8)	80 (64.0)
More/Increase	37 (12.8)	45 (30.4)	28 (22.4)
Change from previous?	-	*p*-val = 0.001	*p*-val = 0.324
Daily fruit intake	*n* = 211	*n* = 148	*n* = 125
Less/Decrease	36 (17.1)	7 (4.7)	13 (10.4)
No Change	148 (70.1)	95 (64.2)	84 (67.2)
More/Increase	27 (12.8)	46 (31.1)	28 (22.4)
Change from previous?	-	*p*-val < 0.001	*p*-val = 0.196

**Table 2 nutrients-14-04865-t002:** Self-reported health and weight changes: prior to pandemic (PP0) to baseline (M0).

**Do you think your general health changed during the height of the COVID-19 pandemic in your area? (*n* = 232)**	***n* (%)**
Yes, I think my health declined	119 (51.3)
No change	90 (38.8)
Yes, I think my health improved	23 (9.9)
**Self-reported weight change during height of pandemic (*n* = 226)**	***n* (%)**
>10 pounds lost	16 (7.1)
6–10 pounds lost	16 (7.1)
1–5 pounds lost	21 (9.3)
No weight change	40 (17.7)
1–5 pounds gained	44 (19.5)
6–10 pounds gained	39 (17.3)
>10 pounds gained	50 (22.1)

**Table 3 nutrients-14-04865-t003:** Mean (standard deviation) weights over time, by BMI category.

BMI	PP0 (*n* = 226)	M0 (*n* = 226)	M6 (*n* = 148)	M12 (*n* = 122)
Underweight/Normal	133.2 (16.2)	136.0 (17.9)	134.6 (17.9)	134.8(19.3)
Overweight	164.5 (15.3)	168.8 (17.6)	167.1 (19.4)	165.4 (23.3)
Obesity	226.8 (43.5)	228.3 (45.7)	224.8 (51.4)	224.4 (52.8)
All	163.6 (43.8)	166.6 (44.6)	164.5 (44.5)	165.1 (46.4)

**Table 4 nutrients-14-04865-t004:** Mean (SD) of number of exercise sessions by intensity.

Exercise Level	Prior (*n* = 218)	Height (*n* = 218)	M6 (*n* = 148)	M12 (*n* = 124)
Mild	3.55 (2.60)	3.13 (2.62)	5.17 (3.29)	4.84 (3.09)
Moderate	2.33 (2.08)	2.14 (2.47)	3.84 (2.61)	3.13 (2.28)
Strenuous	2.12 (2.09)	1.54 (2.12)	2.99 (2.17)	2.69 (2.29)
Total	8.00 (5.09)	6.80 (5.67)	12.00 (6.37)	10.67 (5.78)
Godin Score	41.40 (28.22)	33.90 (30.92)	61.64 (33.62)	54.40 (32.13)

**Table 5 nutrients-14-04865-t005:** Psychological factors at M0 and M12: depression (PHQ-2), anxiety (GAD-7), insomnia (ISI).

Psychological Variable	Category	M0	M12
Depression (PHQ-2) (M0: *n* = 234, M12: *n* = 124)	None (0)	69 (29.5)	51 (41.1)
	Mild (1)	42 (17.9)	20 (16.1)
	Moderate (2)	64 (27.4)	34 (27.4)
	Severe (3–6)	59 (25.2)	19 (15.3)
Anxiety (GAD-7) (M0: *n* = 234, M12: *n* = 124)	None (0–4)	69 (29.5)	58 (46.8)
	Mild (5–9)	101 (43.2)	38 (30.6)
	Moderate (10–14)	42 (17.9)	16 (12.9)
	Severe (15–21)	22 (9.4)	12 (9.6)
Insomnia (ISI) (M0: *n* = 229, M12: *n* = 124)	Normal (0–7)	63 (27.5)	26 (21.0)
	Subthreshold (8–14)	99 (43.2)	57 (46.0)
	Moderate (15–21)	58 (25.3)	30 (24.2)
	Severe (22–28)	9 (3.9)	11 (8.9)

## Data Availability

The data presented in this study are available on request from the corresponding author. The data are not publicly available due to privacy.

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
