# Peer review of "A Year in the Life of U.S. Frontline Health Care Workers: Impact of COVID-19 on Weight Change, Physical Activity, Lifestyle Habits, and Psychological Factors"

_nutrients, 2022, doi:10.3390/nu14224865_

Round 1

Reviewer 1 Report

Dear Authors,
The topic of the article is very interesting and "promising", but its performance raises many questions. I am not sure what the authors wanted to achieve and demonstrate as a result of the analysis brought. I will refer to the individual sections of the article below.

Abstract
This part of the text needs changes after the article was revised. The most important parts of the abstract are the purpose of the research (consistent with the title) and conclusions consistent with the purpose, and this condition has not been fully met.

Introduction
In the chapter, I would ask you to add brief information on the course of the pandemic in the US, with particular emphasis on the periods of the study (M0, M6 and M12) - this would make it easier to relate the results obtained to the state of the pandemic situation.
It is also necessary to clearly formulate the purpose of the study and present the research questions and/or hypotheses.

Materials and methods
Subsections 2.1., 2.2., 2.3. do not raise major objections, doubts about the statistical analyses assumed by the authors (2.4.) will be presented in the section of the review on the description of the results.

 Results
I suggest moving the first part of the description of results to subsection 2.2. (Participants)
The title of the work and the introduction suggest that the subject of analysis will be changes in the health status, physical activity, mental state of medical workers at different stages of the pandemic, such a way of approaching the research material would be very interesting and would lead to conclusions that are interesting from a cognitive and application point of view.
3.1.
I don't quite understand the purpose and sense of taking into account changes in dietary habits of BMI during the analyses (although it is not wrong), as well as focusing only on PP0 and M0 periods for comparisons. In my opinion, it would have been more interesting to determine changes in all the periods in which the study was conducted. I also consider it necessary to provide detailed results of statistical analyses (e.g., statistical significance of differences) in the table. In summary, it would be interesting to determine how the dietary behavior of the surveyed group of respondents changed at different stages of the pandemic.
3.2.
The caveats to this section are analogous to those of section 3.1. except for the inclusion of BMI, which, in the case of weight changes, seems reasonable.
3.3.
The strength of this subsection is the inclusion of all measurement periods, although the table still includes descriptive statistics without providing the most relevant data on the variation of results, which is the basis for inference. I consider the analyses of the relationship between physical activity and weight change to be unnecessary, in the context of the title of the work (and, one guesses, its purpose).

3.4.
In the case of psychological symptoms (depression, anxiety and insomnia), it would also be advisable to compare results from all study periods and identify differences. The search for the relationship between the intensity of psychological symptoms and physical activity and BMI is a topic for another article. I also emphasize the need to present in tables (and not only in the description of results) the results of statistical analyses.

Discussion
The discussion of the results is quite poor. It cites less than ten items of literature, which is a negligible number, considering the multitude of scientific texts on the impact of the pandemic on dietary behavior, physical activity and mental state of the populations of various countries. Assuming that the authors would have heeded my earlier suggestions, this section of the article would have to assess whether the results obtained are consistent with those of other studies on medical workers, and whether and how much the changes in the behavior of this subpopulation are different from those of "ordinary" people during a pandemic, as well as discuss the reasons for these possible differences.
Conclusions
 In this chapter I would expect a presentation of Key findings (it does not seem necessary to include them in a table), as well as general conclusions of the study (also of an applied nature).
I hope that my comments will be helpful after proofreading the text. I believe that the authors have interesting empirical material, which can form the basis for the preparation of a valuable article.

Author Response

Thank you for your helpful feedback.  Responses to your edits and suggestions are attached.  We hope we have been able to fulfill your requests. 

Reviewer #1 Comments

Author Response

Dear Authors,
The topic of the article is very interesting and "promising", but its performance raises many questions. I am not sure what the authors wanted to achieve and demonstrate as a result of the analysis brought. I will refer to the individual sections of the article below.

We thank and appreciate Reviewer #1 for the very helpful feedback on this manuscript.

Abstract

This part of the text needs changes after the article was revised. The most important parts of the abstract are the purpose of the research (consistent with the title) and conclusions consistent with the purpose, and this condition has not been fully met.

The title has been updated to remove the word health and added in nutrition and lifestyle habits which may be more inclusive terminology for this study.

We also believe the terminology nutrition is more inclusive of eating behaviors and weight therefore the abstract is currently reflective of this. 

Introduction

·         In the chapter, I would ask you to add brief information on the course of the pandemic in the US, with particular emphasis on the periods of the study (M0, M6 and M12) - this would make it easier to relate the results obtained to the state of the pandemic situation.

·         It is also necessary to clearly formulate the purpose of the study and present the research questions and/or hypotheses.

Thank you for your recommendation. Brief information related to COVID-19 during the study periods of M0 through M12 was added to introduction section. Lines 33-38.

A purpose/hypothesis statement was added for clarification. Lines 85-91.

Results

·         I suggest moving the first part of the description of results to subsection 2.2. (Participants)

Thank you for your suggestion. The first 3 sentences in the Results section were moved to section 2.2, Participants.

3.1.

I don't quite understand the purpose and sense of taking into account changes in dietary habits of BMI during the analyses (although it is not wrong), as well as focusing only on PP0 and M0 periods for comparisons. In my opinion, it would have been more interesting to determine changes in all the periods in which the study was conducted. I also consider it necessary to provide detailed results of statistical analyses (e.g., statistical significance of differences) in the table. In summary, it would be interesting to determine how the dietary behavior of the surveyed group of respondents changed at different stages of the pandemic.

Additional data on eating pattern changes (3.1) at different pandemic stages have been added and the BMI comparisons have been removed. Now data includes timepoints M0, M6, and M12 as well as statistical differences where warranted.   Lines 168-193. Table 2 has been updated accordingly.

3.2.

The caveats to this section are analogous to those of section 3.1. except for the inclusion of BMI, which, in the case of weight changes, seems reasonable

Thank you for this observation. No additional changes have been made to this section.

3.3.

The strength of this subsection is the inclusion of all measurement periods, although the table still includes descriptive statistics without providing the most relevant data on the variation of results, which is the basis for inference. I consider the analyses of the relationship between physical activity and weight change to be unnecessary, in the context of the title of the work (and, one guesses, its purpose).

Thank you for this comment. While the table includes the descriptives, lines 247-255 include the time points' comparisons and their statistical significance.

We appreciate your insight regarding PA and relevant weight changes and have decided to continue to include and report these findings.

3.4.

In the case of psychological symptoms (depression, anxiety and insomnia), it would also be advisable to compare results from all study periods and identify differences. The search for the relationship between the intensity of psychological symptoms and physical activity and BMI is a topic for another article. I also emphasize the need to present in tables (and not only in the description of results) the results of statistical analyses.

Thank you for your recommendations. Additional data and psychological factors (3.4) at different stages of the pandemic have been added to include the only other time point which collected this data (M12 ) as well as the comparisons and statistical analysis requested. Table 6 has been revised accordingly as well as lines 291-298; 330-335; 368-369.

BMI and PA data was left in this article as these findings may be of interest to this readership.

Discussion

The discussion of the results is quite poor. It cites less than ten items of literature, which is a negligible number, considering the multitude of scientific texts on the impact of the pandemic on dietary behavior, physical activity and mental state of the populations of various countries. Assuming that the authors would have heeded my earlier suggestions, this section of the article would have to assess whether the results obtained are consistent with those of other studies on medical workers, and whether and how much the changes in the behavior of this subpopulation are different from those of "ordinary" people during a pandemic, as well as discuss the reasons for these possible differences.

Extensive revisions were applied to the discussion section to satisfy these concerns as well as respond to the new findings that were included as part of these revisions. Additional references have also been included.

Conclusions

 In this chapter I would expect a presentation of Key findings (it does not seem necessary to include them in a table), as well as general conclusions of the study (also of an applied nature).

The conclusion section has been updated accordingly.

Reviewer 2 Report

Review of Manuscript ID: nutrients-1975119 titled "A Year in the Life of U.S. Frontline Health Care Workers: Impact of COVID-19 on Health and Lifestyle Habits" by Oliver et al. 

The authors investigated the effect of the COVID-19 pandemic on healthcare workers habits. Data on weight changes, eating behaviors, physical activity, and psychological factors were investigated. The main findings were that the COVID-19 pandemic had immediate effects on the body weight and physical activity levels but this were normalized one year later.  

Well written manuscript with relevant data. I have no major criticism only have some minor comments.

The manuscript is too long (booth the introduction, results and the discussion)

The authors should use the metric system for reporting, weight should not be reported in pounds (you are not consistent because you use BMI)

There are unnecessary number of decimals for p-values and R-values

Author Response

Thank you for your helpful feedback.  Responses to your edits and suggestions are attached.  We hope we have been able to fulfill your requests. 

Reviewer #2 Comments

Author Response

Well written manuscript with relevant data. I have no major criticism only have some minor comments.

Thank you for taking the time to review our manuscript as well as your suggestions for improvement!

The manuscript is too long (both the introduction, results and the discussion)

Thank you for your comment.

With the longitudinal nature of this work, we do understand that it is lengthy but hopefully an important contribution to the literature.

The authors should use the metric system for reporting, weight should not be reported in pounds (you are not consistent because you use BMI)

Thank for your this suggestion. The authors considered your suggestion, but concluded that since the survey was administered originally to U.S. health care workers using the standard system, reporting the results in standard form in tables and text seemed appropriate. Although BMI is reported in metric form, this is universally accepted in literature using metric and standard systems.

There are unnecessary number of decimals for p-values and R-values

We have adjusted the r-values on lines 229-234. Additionally, the p-values on lines 247, 308, 332, 341 have been reduced and rounded to three digits.

Noted: English language and style are fine/minor spell check required

Thank you for your observation. We have conducted additional manuscript proofreading.

Round 2

Reviewer 1 Report

Dear Authors,

The changes made are undoubtedly beneficial from the point of view of the structure of the text, nevertheless it is still chaotic.  I have the impression that the Authors raise too many threads in the description and for this reason they find it difficult to keep the narrative in order.

Once again, I suggest posing research questions in the introduction (perhaps divided into main and specific ones) and organizing both the presentation of results and the discussion around these questions, leaving out side threads. The key findings included in the individual subsections in the Results section should also relate to the research questions (I see no need to include them additionally in the Discussion section). I also do not understand the intention to present in the Results chapter the characteristics of the M0 population with particular reference to BMI . Rather, the population characteristics of all studies should be included as supplementary materials.  The changes made in the Conclusions chapter are also not satisfactory - I would have expected a link between the conclusions drawn and the research questions and key findings formulated in the description of the results. In addition, these conclusions (in an abbreviated form) should be included in the abstract.

I would very much like to ask you to reconsider the text and, above all, to formulate research questions that will be the axis of the presentation and discussion of the results, which is still too superficial, especially with regard to Psychological Symptoms.

The authors repeatedly emphasize that research on HCWs has not been done before, but in my opinion it would be possible and interesting to relate the results obtained to the results of research on "ordinary people".

I do not understand what purpose is served by referring in the discussion to results not presented in the text (line 380). 

For the most part, the results obtained are very interesting and worthy of publication, so I strongly encourage the Authors to think about the text (suggesting at least the title), extract the most important issues and focus on discussing them. Perhaps other topics raised can inspire another publication.

Author Response

Please see attached!

Thank you for the opportunity to revise and resubmit.

Reviewer #1 Comments

Author Response

The changes made are undoubtedly beneficial from the point of view of the structure of the text, nevertheless it is still chaotic.  I have the impression that the Authors raise too many threads in the description and for this reason they find it difficult to keep the narrative in order.

In response to this request we have added research questions to the introduction section. The results and discussion section follow this outline throughout the paper. We hope the addition of the research question enhances the flow and serves as a guide to the reader throughout the paper.

Once again, I suggest posing research questions in the introduction (perhaps divided into main and specific ones) and organizing both the presentation of results and the discussion around these questions, leaving out side threads.

In response to this request we have added research questions to the introduction section. The presentation of the results and discussion section follow these research questions. We have deleted any text that did not align with these research questions.

The key findings included in the individual subsections in the Results section should also relate to the research questions (I see no need to include them additionally in the Discussion section).

The key findings as well as Table 7 have been removed from this paper. Table 7 can now be found in Appendix A listed as Table A2.

I also do not understand the intention to present in the Results chapter the characteristics of the M0 population with particular reference to BMI . Rather, the population characteristics of all studies should be included as supplementary materials.  

Thank you for bringing this descprenacy to our attention. We have revised the results section to include 3.1 Descriptive analysis which includes a summary of all baseline data. The original Table 1 with all general demographics has been removed from the text and changed to table A1, in Appendix A.

The changes made in the Conclusions chapter are also not satisfactory - I would have expected a link between the conclusions drawn and the research questions and key findings formulated in the description of the results. In addition, these conclusions (in an abbreviated form) should be included in the abstract.

The abstract has been updated to include all of the key findings.  The key findings (and table) have been removed from the discussion to avoid redundancy. The discussion and conclusions have been updated to reflect the order of the research questions and their findings. We hope these revisions have addressed your concerns.

I would very much like to ask you to reconsider the text and, above all, to formulate research questions that will be the axis of the presentation and discussion of the results, which is still too superficial, especially with regard to Psychological Symptoms.

Thank you for this suggestion. Research questions have been added to the introduction section. The results and subheadings within the results reflect these research questions and their findings. We hope this new outline and formatting will aid the article's readability and guide the reader through the research study's findings. Findings that were not relevant to these research questions have been removed (i.e. BMI and psychological factors).

The authors repeatedly emphasize that research on HCWs has not been done before, but in my opinion it would be possible and interesting to relate the results obtained to the results of research on "ordinary people".

Thank you for this comment. The authors believe this has been previously addressed as noted in lines 434-439; 461-469; 476-481; 487-493; 506-508; 552-555; 610-614.

Additional edits were made in section 4.4.   Please see lines 451-545; 555-560.

I do not understand what purpose is served by referring in the discussion to results not presented in the text (line 380). 

Thank you for this comment. Lines 451-456 have been removed.

For the most part, the results obtained are very interesting and worthy of publication, so I strongly encourage the Authors to think about the text (suggesting at least the title), extract the most important issues and focus on discussing them. Perhaps other topics raised can inspire another publication.

Thank you for this feedback. The title and abstract have been revised and research questions  have been added in efforts to clarify these research findings and best serve the reader to highlight the key issues and findings of this research.
